# Enhancing the Stability of Bacteriophages Using Physical, Chemical, and Nano-Based Approaches: A Review

**DOI:** 10.3390/pharmaceutics14091936

**Published:** 2022-09-13

**Authors:** Mateusz Wdowiak, Jan Paczesny, Sada Raza

**Affiliations:** Institute of Physical Chemistry, Polish Academy of Sciences, Kasprzaka 44/52, 01-224 Warsaw, Poland

**Keywords:** antibiotic resistance, phage therapy, phage stabilization, lyophilization, encapsulation, nano-assisted stabilization, polymers

## Abstract

Phages are efficient in diagnosing, treating, and preventing various diseases, and as sensing elements in biosensors. Phage display alone has gained attention over the past decade, especially in pharmaceuticals. Bacteriophages have also found importance in research aiming to fight viruses and in the consequent formulation of antiviral agents and vaccines. All these applications require control over the stability of virions. Phages are considered resistant to various harsh conditions. However, stability-determining parameters are usually the only additional factors in phage-related applications. Phages face instability and activity loss when preserved for extended periods. Sudden environmental changes, including exposure to UV light, temperature, pH, and salt concentration, also lead to a phage titer fall. This review describes various formulations that impart stability to phage stocks, mainly focusing on polymer-based stabilization, encapsulation, lyophilization, and nano-assisted solutions.

## 1. Introduction

Bacteriophages are viruses infecting bacteria. The name is derived from “bacteria” and the Greek φαγεῖν (phagein), meaning “to devour”. Bacteriophages are obligate parasites that annex the host’s molecular machinery to complete their life cycle. Hundreds of new virions are folded inside a single bacterial cell. In most cases, bacteria are disrupted, and phages are released (lytic cycle). Chronic phages, e.g., filamentous phages f1, fd, or M13, do not lyse their host cells, but progeny virions are secreted continuously [1]. Phages that can undergo only the lytic cycle are called virulent, and those which can undergo lytic and lysogenic (latent) cycles are called temperate [2,3]. In the lysogenic cycle, the viral genome integrates into the chromosomes of bacteria, remains latent, and replicates with the host [4]. Such an integrated state is known as a prophage [5]. Stressors, e.g., chemicals, UV radiation, or damage to the host DNA, can initiate the lytic cycle to escape the endangered host [6]. The natural ability of most bacteriophages to destroy bacteria makes them a great candidate to fight multi-drug-resistant infections or even replace antibiotics in the post-antibiotic era.

The average size of a virion (single phage) is around 50 nm to 200 nm. However, the largest bacteriophages are more than 800 nm [7]. In ocean water, bacteriophages with tails of about 1800 nm have occasionally been observed [8]. The vast majority of all known bacteriophages (above 95% [9,10]) share a typical structure design, i.e., genetic information (dsDNA) is stored in an icosahedral capsid to which a spike-tail with fibers is attached. The length and stiffness of the tail spike depend on whether they belong to the family *Straboviridae* (long and contractile tail, e.g., T4), the *Caudoviricetes* class (long, noncontractile tail, e.g., λ phage), or the family *Autographiviridae* (short, noncontractile tails, e.g., T7). Tail fibers are attached to the tail and have a substantial positive charge. The head has a significant negative charge [11]. Such asymmetry results in a permanent dipole moment of the virion. Much less common than *Caudoviricetes* class phages are filamentous (e.g., M13, fd) or nearly spherical (isometric) phages (e.g., MS2).

Only recognizing a proper and viable host assures the multiplication of virions and the completion of the phage life cycle. Thus, very often, a multi-step “identification” process is used. In the case of *Caudoviricetes*, initial recognition is based on electrostatic interactions—positively charged tail fibers are attracted to the negatively charged surface of bacteria. The following steps of host recognition utilize specific receptor-binding proteins, which discriminate the appropriate host. This selectivity is the basis for the utilization of phages in sensing.

The phage display method boosts the importance of bacteriophages. The possibility of directly studying the link between genotype and phenotype has allowed for numerous applications. Phage display is quite similar to the immunodetection assay but, instead of relying on antibodies, it relies on phages as recognition elements. Briefly, a library of genetically modified phages expressing different peptides on their capsids is prepared. Then, the phage library is exposed to the molecule (embedded on the surface) to determine specifically binding peptides [12].

Confidence in phage therapy is re-emerging in the pharmaceutical world. The FDA approved the use of phages in critically ill COVID patients with secondary bacterial infections in 2020. Several phage-based products emerged shortly after, for example, GangaGen, ContraFect, Phagelux, and Phagomed, to fight S. aureus in India, the USA, China, and Austria, respectively [13]. In addition to phage therapy and biocontrol applications, sensing, and phage display, bacteriophages are also used in drug delivery [14], material science, and nanotechnology [15] virus-related studies (e.g., vaccines) [16]. In all of these cases, phages are our allies. Increasing the stability of formulations is crucial for the successful development of phage-based applications. Here, we review the attempts to achieve phage stabilization.

Approximately 13% of deaths are related to bacterial diseases [17]. Bacteria are also involved in specific types of cancers [18] and metabolic disorders [19]. The World Health Organization (WHO) is alarmed that 4.1 million patients are affected by healthcare-related illnesses each year in Europe [20]. In the USA, nosocomial infections cause 100,000 deaths yearly [21].

Foodborne illnesses remain a significant cause of worldwide death despite many advances in pathogen surveillance and food sanitation methods. According to the WHO, 600 million foodborne infections occurred in 2010, resulting in over 400,000 deaths. Besides being a huge social burden, it is also a massive drain on the economy of nations. The average incident is estimated to cost around USD ~1500 per person [22]. Moreover, microbial contamination is the major cause of food spoilage, resulting in the loss of 25% of food produced yearly [23].

Being natural antibacterial agents, bacteriophages are considered an alternative to antibiotics, especially at the dawn of the antibiotic-resistance era. There are several advantages to using phages to fight bacteria.

Phages can be produced easily and cheaply in large quantities and easily purified. By only infecting a bacteria solution, one can obtain a large number of progeny phages.Bacteriophages are considered non-toxic to eukaryotes because structural elements of the virion cannot bind to eukaryotic cells [24]. There are, however, specific examples of phage internalization by eukaryotic cells. Lehti et al. demonstrated the penetration of eukaryotic cells by the *E. coli* phage PK1A2 in vitro [25]. The virus remained in the cells for up to 24 h but did not affect cell viability.The lysis of bacteria resulting from phage infection supports the inflammatory response against bacteria [26]. Therefore, phage therapies directly eliminate bacteria cells and indirectly activate humans’ immune systems.Phages undergo evolution, and thus, they remain effective against bacteria [27]. Despite bacteria developing countermeasures against phages [28], phages are also adapting [29].

Bacteriophages have been proposed for medical use since the early 1900s [4,30]. Due to a poor understanding of phages, issues related to the stability of formulations, and, later, the appearance of antibiotics, phage therapies were neglected for decades [31]. The spread of drug-resistant superbugs and the lack of new medicines [32,33] has caused a renaissance of bacteriophage-based antimicrobials [34]. Nowadays, phages are used to fight infections that do not respond to conventional antibiotics [35]. In some countries, e.g., Russia and Georgia, phage products are available over the counter, even without a prescription [36,37]. In western countries, phage-based methods are advancing in clinical trials for the treatment of inner-ear infections [38], typhoid [39], and burn wound infections (*Phagoburn* project) [40]. The first clinical trial in the USA was already approved in 2019 [41].

Phage administration without any stabilizing additives triggers an inflammatory response as a result of releasing pathogen-associated molecular patterns (PAMPs) from lysed bacteria (i.e., membrane proteins, LPS, etc.) [26]. However, phages also impact immunity directly, as they modulate the innate and adaptive immune response through phagocytosis, the cytokine response, and antibody production [24]. Anti-phage antibodies are one of the most significant factors limiting the therapeutic effectiveness of phage therapy [42,43]. The antibodies usually recognize phage receptors as antigens, and thus, binding to them *de facto* makes phages incapable of infecting bacterial cells [3].

Phages are also tested for biocontrol applications, e.g., in the food industry and agriculture. Phage biocontrol is increasingly accepted as green and natural technology for the specific targeting of pathogens [44]. Phages are a viable alternative to antibiotics [45]. Phages were used to protect dairy products [46], fruits [47], vegetables [48], meat [49], and fish [50]. In 2020, among over fifty commercial entities offering phage-derived products, fifteen were focused on biocontrol [51]. There are more than ten products approved for food safety applications [52]. Developments in the utilization of phages as antimicrobial agents in plant and animal agriculture are summarized in a review by Svircev et al. [53].

Among the many bacteriophages, some examples are considered good surrogates for studies on eukaryotic, often dangerous, viruses. The most common examples are MS2 [54], Phi6 [55,56], PhiX174 [57], and QBeta [58]. For instance, MS2 phages were employed as a model agent for inactivation studies due to their resistance to UV and the resemblance between their inactivation profile and some enteric viruses, such as poliomyelitis virus [54,59]. Phi6 was used as a surrogate of Sars-Cov-2 in studies on survival in evaporated saliva droplets on glass surfaces [55] and tests on surface disinfectants [60]. PhiX174 and QBeta were used for studies on virus deactivation [57,58,61]. Figure 1 shows the similarity between Phi6 phage and SARS-CoV-2 on SEM images.

As such, methods developed for phage stabilization might be applicable to the stabilization of other viruses. This is still important, as bacteriophages can be used to formulate vaccines. Depending on the particular type of vaccine, phages can either carry the genomes of other viruses, affinity bind eukaryotic virus antigens on the capsid’s surface [16,62], or present virus antigens to select its targeting factors via phage display [63]. Moreover, as phages are surrogates of eukaryotic viruses [64], different strategies of vaccine stabilization can be examined on phage models.

**Figure 1 pharmaceutics-14-01936-f001:**
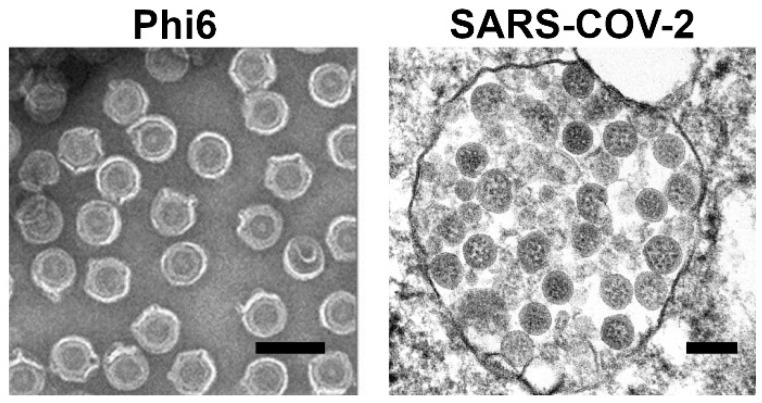
The comparison of enveloped phage Phi6 [65] and SARS-CoV-2 virus [66] using scanning electron microscopy imaging. The scale bar is 100 nm. Phi6 image was adapted from Block et al., based on CC BY 4.0 License [65]. SARS-CoV-2 image was adapted from Goldsmith et al. [66] based on Elsevier License No. 5375361146341.

The major drawback of using phages as antimicrobials is the stability of virions. The primary criteria are virulence, selectivity, host range, ease of manipulation, and modification. Even though phages are capable of retaining their activity after exposure to stress factors (including high temperatures [67], pH [68], and organic solvents [69,70]), this resistance is only an additional factor in the search for valuable medical purposes. Stability-determining factors were described in the review by Jończyk-Matysiak et al. [71]. The authors focused on the effects of chemicals and physical factors, including pH, temperature change (freezing, heat), and UV-light.

The matter of phage suspension stability is not to be underestimated. Improper preparation of such suspension can affect the effectiveness of the phage therapy approach. For instance, in the *Phagoburn* project, storing bacteriophages in unsuitable conditions dramatically decreased phage titer a thousand-fold within just fifteen days. Because of this, patients received lower dosages of phages (10^2^ PFU/mL daily), about 5 log below the required rate of infection (ROI) [40].

With the shift in attention towards phage-based pharmaceutics, the precise nature and uniformity of bacteriophages are gaining increased appreciation. The preliminary testing of phage therapy did not concern the stability of phages; however, commercial circulation would require stable concentrations and known dosages. Phages often lose activity during formulation and storage. The most common factors affecting the viability of phages (and phage proteins) include exposure to organic solvents, high temperatures, pH, and ionic strength [72].

## 2. Polymer-Based Stabilization

A polymer is a substance or material of large molecular mass composed of repeating subunits (*Encyclopedia Britannica*). Because of a large number of subunits, removing one of them does not change the properties of the entire molecule. Polymers can be divided into natural (proteins, nucleic acids, and polysaccharides) and synthetic (polyethylene, polytetrafluoroethylene, and polypropylene) [73]. Although polymers’ usage for phages and virus stabilization is in the early stages, some interesting applications have already been reported.

The effect of sugars as stabilizers against protein unfolding is known [74,75,76,77,78]. Because of this effect, sucrose has been used as a cryo-stabilizer for freeze-dried vaccines. Yet, it has been shown that sucrose-induced stabilization of the viral capsid proteins alone does not necessarily lead to virus capsid stabilization [74]. More importantly, the protein stabilization effect is broadly observed with many sugars [79,80,81,82], but only sucrose and trehalose seem to be effective in stabilizing viral capsids [79,83]. Sucrose is known to enhance protein–protein binding at molar concentrations by modifying protein hydration properties [75], and thus, it is effective in the stabilization of vaccines [79,83,84,85]. Other carbohydrates, such as agar or alginate, can also be used for the stabilization of phage particles. Balcão et al. proposed fifteen polymers with different concentrations of sodium alginate and calcium solution for storing bacteriophage DSM JG004, targeting *P. aeruginosa*. Such hydrogels were effective in the elimination of bacteria embedded on the surfaces of gel beads. The authors, however, did not provide quantitative values of these antibacterial properties. [86]. Leung et al. proposed the usage of pullulan and trehalose films for the protection of the LISTEX P100 phage, targeting *Listeria monocytogenes.* While pullulan itself was not effective for phage stabilization after drying, the combination of pullulan and trehalose as a stabilizing matrix allowed the maintenance of about 7 log active phages after 60 days (from the initial concentration of 10 log). In comparison, in the trehalose matrix, only 3 log phages remained active after the same amount of time [87].

Polyacrylamide can be used to stabilize phage particles for strictly research purposes, such as protein nuclear magnetic resonance spectroscopy. Trempe et al. reported the stabilization of a filamentous Pf1 phage using 5% polyacrylamide as a polymer-stabilized liquid crystal (PSLC). This approach allowed for measurements of dipolar couplings with a single sample and, therefore, a more accurate analysis of protein structure based on the comparison of theoretical and experimental tensor parameters [88].

PEGylation is the attachment of polyethylene glycol (PEG) groups to the target molecule, commonly used in food and drug formulation to increase molecules’ stability. PEG is biocompatible and reduces the immunogenicity of the molecule, but it is non-degradable [89]. PEG is known to affect protein–protein interactions by changing their hydration, similarly to sucrose [90]. Since phage capsids are made exclusively of proteins, PEGylation was found to be suitable as a phage stabilizer. This approach uses PEG polymer conjugate with phage proteins [91]. Kim et al. reported that two bacteriophages—Felix-O1 and A511—when modified with monomethoxypolyethylene glycol (mPEG) could be present for longer within the bloodstream. After 24 h, the amount of non-modified phages was 2 log less than mPEG-modified. However, their immunogenicity was increased, which was suggested by the larger amount of IgG and IgM antibodies 12 days after the injection [92]. PEGylated phage lysins are currently considered to be more promising [93]. In comparison, the PEGylation of phage endolysins caused the loss of their bactericidal properties in vitro [94].

Polymers might also have an indirect effect on phages. Richter et al. explained the phenomenon of phage titer loss when phages are stored in plastic (mainly polypropylene) labware [95]. The authors observed that the effect depends on the hydrophobicity of the inner walls of the plastic containers and tubes. For high-enough wetting angles, the system minimizes energy by limiting the contact between hydrophobic plastic and water molecules. The surface nanoporography also appeared significant for the absorption process [96]. In the case of phages, it resulted in dramatic changes in the bulk titer. To overcome this effect, Richter et al. proposed the addition of Tween-20 and plasma treatment [95]. Later, O’Connel et al. used pre-incubation with bovine serum albumin [96]. Additionally, an inorganic polymer nanocomposite (gold nanoparticles embedded in polyoxoborates) can be used to cover plastic container surfaces. Such a nanocomposite not only prevents the uncontrolled absorption of phages, but also has antibacterial and antifungal properties. [97]. This, indirectly, allows for more extended storage of liquid phage stocks. Both the phage absorption of plastic surfaces and its prevention are presented in Figure 2.

## 3. Encapsulation

Polymers are also used as protective matrices in which phages are embedded. Encapsulation allows for better stability and modulates the long-term release of active phage particles [98]. Various techniques are used for encapsulation, for example, emulsion, polymerization, spray-drying, and extrusion dripping [71]. The encapsulation of bacteriophages for preservation and administration must consider the possible stresses encountered by phages during and after the process. It is crucial to ensure that the morphology of encapsulated particles is maintained and that they do not aggregate or uncontrollably fuse with the surface of interest [72]. The bacteriophage strain of interest defines the encapsulation conditions; it is difficult to develop methods that universally apply to phages [71]. The encapsulation of phages also facilitates the transportation of samples at prescribed temperatures for long durations. For example, Menendez et al. showed that microencapsulated phages could be maintained at 20 °C for 2 months [99]. Dry encapsulation also has an advantage over other liquid-based formulations, especially for transporting phages and maintaining a long shelf-life [100].

The applications of encapsulation also include phage cocktails. For example, a cocktail of three phages against *Salmonella* was encapsulated within alginate microparticles associated with calcium carbonate to prolong their gut residence time [101,102]. Pacios-Michelena et al. combined alginate/chitosan-encapsulated phages with polyphenolic extracts, increasing the phage surveillance during UV exposure from about 5 min to at least 25 min [103].

Malik et al. listed the types of polymers used for phage encapsulation as agarose, alginate, chitosan, pectin, whey protein, gelled milk protein, hyaluronic acid methacrylate, hydroxypropyl methylcellulose poly(N-isopropyl acrylamide), poly(DL-lactide: glycolide), polyesteramide, polyvinyl pyrrolidone, polyethylene oxide/polyvinyl alcohol, cellulose diacetate, and polymethyl methacrylate [72]. Alginate is often used to encapsulate phages for applications that require exposure to acidic media [102]. Tang et al. improved the release of phage particles from alginate microspheres by using whey protein [71]. Alginate capsules can also be developed to immobilize phages and their storage at lower temperatures [99]. Furthermore, bead encapsulation in chitosan–alginate multilayers can enhance the stability of phages even under harsh conditions such as in the intestine [104].

Liposomes are biocompatible nanoparticles composed of a lipid bilayer surrounding therapeutic cargo. This approach is promising in achieving directed delivery while surviving the extreme conditions of the stomach and intestine when administered orally [98]. Liposome-encapsulated phages were used to target *Salmonella* [71]. Pharmaceutical formulations in the form of gastro-resistant microparticles encapsulating anti-*Salmonella* phages have been produced to maintain elevated levels of phage particles in the gut [102]. Nanoparticles of cationic liposomes can also be designed to encapsulate anti-*Salmonella* phages, protecting them against the acidic conditions of the intestine [102]. This technique is popularly adopted to immobilize phage cocktails to treat burn infections [105]. However, liposome-encapsulated phages appeared to be less effective in comparison to phages alone. This fact led researchers to explore hydrogels as scaffolds (primarily polymers) for phage delivery [106].

While aiming for perfection, modifications in the approach to encapsulate phages are gaining popularity. For example, microfluidic devices are used to produce calcium alginate capsules containing bacteriophages [107]. These capsules find applications in the sanitization of food surfaces. There is a constant need to improve the technique to avoid the exclusion of encapsulated phages by our immune systems and the cleavage of capsid proteins by proteases in the gut. The efficacy of encapsulated phages is presented in Figure 3.

## 4. Lyophilization

Lyophilization (dehydration process), first used for food storage, has become a very commonly used method for phage stabilization and long-term storage. At first, lyophilization involved freezing the phage stock, lowering the pressure, and removing the water (so-called freeze-drying) [109]. Now, there is a branch of similar methods, including: (i) spray-drying, where the concentrated liquid is atomized and exposed to the hot air—causing evaporation of the water—and then, dried and separated [109,110]; (ii) hot-air-drying, where the sample is pre-treated with ethyl oleate and potassium carbonate solutions, and then, exposed to a temperature of about 50–60 °C [111,112]; and (iii) drum-drying, where hot-air-drying is enhanced by placing the sample in a rotating drum to increase the heat transfer [113]. The differences in phage formulations with freeze-drying, spray-drying, and electrospray are presented in Figure 4.

For phage storage, freeze-drying and spray-drying methods are the most important. Freeze-drying is, relatively, the cheapest method for preparing phage powders. One of the first attempts was the freeze-drying of mycobacteriophages; when stored in the dark, phage lyophilizates were suitable for over two years [116]. The average loss in titer of such methods was estimated to be about 1 log [117,118]. The activity of phages after freeze-drying can be regulated by the drying time and the number of times they have previously been refrigerated. The authors proved that a long duration of drying (over 150 min) provides three-times-higher survivability of phages during freeze-drying than a short duration (90 and 120 min). Additionally, every subsequent freezing causes the titers to decrease by about 3 log [119]. The efficiency of freeze-drying also depends on the size of the grains of phage powder during the procedure—smaller bead formulation provides a smaller reduction in phage titer than in the case of macro-beads [120].

The storage time can be extended by adding some cryoprotectants. Merabishvili et al. proposed adding 0.5 M sucrose and trehalose solutions to the *Staphylococcus aureus* ISP phage to prolong the storage time up to 37 months, with a titer loss of about 1 log [121]. This observation was confirmed by Dini and Urraza, who also proved how important it is to freeze-dry CA933P phage stocks in proper buffer solutions [122]. Additionally, sugar solutions (sucrose and trehalose) appeared to be much better cryoprotectants than polymers (polyethylene glycol 6000-PEG_6000_) in the M13 phage model. After seven days, the phage titer was 2 log higher in the sucrose and trehalose solution compared to PEG [123]. A similar tendency was described by Puapermpoonsiri et al. [124]. Recently, Petsong et al. proved that the combination of trehalose and whey protein isolate (WPI) allowed the storage of the freeze-dried Salmonella SPT 015 phage, even at room temperature [125]. Other sugars, including lactose or mannose, are not such efficient cryoprotectants, as was shown by Chang et al. [126].

Spray-drying allows for a reduction in the decrease in the phage titer during the procedure [115]. Still, adding carbohydrates to the phage solution is essential for the entire protocol. Vandenheuvel et al. compared the addition of dextran, lactose, and trehalose to phage-titer decrease during the spray-drying of two phages—LUZ19 and Romulus. Trehalose was found to be the most suitable cryoprotectant for the spray-drying protocol [127]. Yet, the virions’ stability after drying strongly depends on the proper crystallization of the trehalose matrix [128]. Leung et al. proposed a mixture of trehalose and leucine in different concentrations to protect two *Pseudomonas* phages—PEV2 and PEV40. They showed that phage powder prepared in the mixture containing 70% trehalose and 30% leucine could be stored for 12 months with a titer decrease of about 0.5 log [129]. Chang et al. proposed mixtures of 80% lactose/20% leucine and 50% lactose/50% leucine for the same purpose on phage PEV20. Such s matrix allowed the storage of phages for 250 days with a loss in titer of about 2 log [130]. In addition to trehalose, Carrigy et al. proposed the usage of trileucine and pullulan as a matrix for phage powders. Depending on the formulation, after 1 month, the phage titer decreased from 0.6 log to 1.9 log [131]. The sensitivity of phages to the drying procedure strongly depends on the particular phage [132]. Moreover, spray-drying can be combined with encapsulation-related methods. That is, a phage that is already encapsulated can be spray-dried, with the possibility of triggered release in response to pH changes, e.g., in the gut [133]. Phage storage and transport can also be facilitated by obtaining dry powders of phage stocks (e.g., phage phiPLA-RODI [99]) containing viable encapsulated phages. Alternatively, metal–phenolic networks can be employed to evade protease cleavage and protect phages until the target is achieved [106].

Freeze-drying and spray-drying were combined into a novel atmospheric spray-freeze-drying (ASFD) [134,135]. Both spray-dried and ASFD phage powders are believed to be a promising medicine for bacterial infections, e.g., caused by *Pseudomonas aeruginosa* [126,130,136,137,138,139,140,141,142,143,144], *Acinetobacter baumannii* [145], *Burkholderia cenocepacia* [146], *Salmonella enterica* [147], *Campylobacter jejuni* [148], or *Mycobacterium tuberculosis* [149]. However, at this moment, such projects are in the stage of optimization [150].

Another long-term method of storing and stabilizing bacteriophages is freezing mature virions within bacterial cells. In this approach, bacteriophages at a proper multiplicity of infection (MOI) are mixed with their host bacterium and incubated for a short time. Next, infected cells are frozen and stored at −80 °C. After reviving and washing, phages are released and can actively infect bacterial cells. Golec et al. proved that this method allows for phage storage with minor or no losses in phage titer for about 10 months, depending on a particular phage [151].

## 5. Nano-Assisted Stabilization

The implementation of nanotechnology in medicine has the potential to solve the stability issue. Nanoparticles can be viewed as vectors for drug solubilization while overcoming sequential biological barriers in the body [152]. Phages can be associated with nanoparticles to remain detectable in the bloodstream for about 24 h longer than in the control [153]. The majority of bacteriophages fulfill the classical definition of nanomaterials (i.e., having one geometrical dimension in a range from 1 nm to 100 nm). Therefore, they integrate well with abiotic nanomaterials, ensuring higher efficiency [154].

Nanoscience plays an essential role in the immobilization of phages; bacteriophages can be chemically or genetically modified to bind strongly to nanomaterials [155]. Gold nanoparticles are most commonly used to stabilize T4-like bacteriophages to detect *E. coli* cells [156]. Additionally, the DNA of *B. anthracis* can be targeted using phage probes modified and stabilized with gold nanoparticles [157]. For more inexpensive applications, silica nanoparticles are popularly employed for their ability to bind to phages [155]. The one-step production of phage-engineered bio-functionalized silicon nanoparticles has also been applied to optical biosensors [158].

In some experiments on vaccine formulations, nanolayers of aluminum oxide were used to stabilize the λ bacteriophage, which ensured the controlled release of antigens in vivo [159]. Vaccines made from eukaryotic viruses may also be stabilized by the addition of nanoparticles. For example, negatively charged gold nanoparticles can improve the storage time of adenovirus [160] (Figure 5).

Apart from imparting stability, nanoparticles have been shown to enhance the functioning of phage-based biosensors by exploring rapid and sensitive approaches [158]. Carbon-based nanomaterials, in particular, have emerged as potential next-generation miniaturized biosensors to obtain susceptible and selective detection [161]. This was displayed when virions were chemisorbed on a glassy carbon electrode decorated with gold nanoparticles, reducing the LOD to 14 CFU/mL within 30 min of incubation [162]. Furthermore, the longevity of such sensors can be increased by using only parts of virions, e.g., receptor-binding proteins (RBPs), instead of the whole bacteriophage [163,164]. Alternatively, MS2 phages can be quickly transported and internalized into bacteria via exposure to Ag and ZnO NPs [165].

The synergistic effect of phages and nanoparticles has been widely used for targeting biofilms and eliminating pathogenic infections [166,167]. For example, the combination of the C3 phage and gold nanoparticles (AuNPs) provided a promising treatment for *P. aeruginosa* planktonic and biofilm states, with high stability under a broad range of temperatures, pHs, and salt concentrations [168]. In other experiments, polyvalent phages were attached to magnetic colloidal nanoparticle clusters (CNCs) to facilitate biofilm penetration [169]. Phage virions may be used as stabilizing agents for synthesizing gold nanoparticles, which have antibacterial and antibiofilm properties [170]. Metallic nanoparticles and bacteriophages impart a synergistic effect against pathogenic bacteria. For example, phage ZCSE6 was combined with ZnO NPs to target *Salmonella enterica* by causing the deformation of biofilms [171]. Alternatively, phage display can be employed to select the most efficient ‘bacteria-recognizing’ peptide. This peptide can be loaded on nanoparticles for an enhanced antibacterial effect [172].

## 6. Conclusions

Vaccination against infectious diseases saves over three million lives yearly. However, some estimations suggest this number could be doubled if all the problems related to proper storage were solved [173]. According to the World Health Organization (WHO) regulations, vaccine stability is defined by three factors: (1) its ability to retain its properties, (2) the duration of retaining its properties, and (3) parameters indicating its stability [174]. Even though recently, mostly mRNA-based vaccines have been desired and examined [175], most vaccines used nowadays contain inactivated pathogens or antigens. To regulate their immunogenicity and minimize the risk of side-effects, pathogen antigens can be displayed on the surface of bacteriophages and are an exciting alternative to ‘traditional’ vaccines [16]. Moreover, the application of microparticles of poly (lactic-co-glycolic acid) (PLGA) for the encapsulation of the modified filamentous phage fd, as a potential anti-cancer therapy, was described. This protocol resulted in the standardized release of bacteriophages during an 8 h period, and was proven to successfully mobilize the release of interleukin 2 (IL-2) by the B3Z hybridoma cell line [176]. Therefore, research on phage stabilization from a long-term perspective may result in a new generation of vaccines that could be used, e.g., in remote parts of the world, where the proper storage of medicines is most problematic [177].

Despite the advantages, research on phages is troublesome due to their instability in phage cocktails and varying phage titers [102]. Table 1 presents the examples of different kinds of formulations to enhance phage stability. When it comes to long-term phage storage, lyophilization and storage inside bacterial cells appear to be most efficient. 

Phage delivery is also often compromised due to its degradation and clearance by the body’s defense mechanisms [178]. Many techniques are adopted to maintain the stability of phages for storage and preservation. Some approaches are met with shortcomings that need further improvements. For example, lyophilization often leads to a distorted morphology which affects the activity of phages [179]. There are discrepancies in the effectiveness of cryoprotectants used to store lyophilized phages. Some experiments have also reported that phages were unstable during 1-year storage at room temperature when they were lyophilized with skim milk and sucrose [180]. Moreover, studies have shown that phage solutions only remain stable up to 126 days after the rehydration of lyophilized phages [181]. The excipients should thus be selected carefully, depending upon the family of the phage of choice. Other studies have found that the encapsulation of phages in liposomes frequently results in undesired aggregation, fusion, or rupture [182]. This also limits the application and further development of such stabilizing methods. Additionally, the nano-based solution for phage stabilization, even though promising, is extremely scarce nowadays, and requires further development. There is also not yet an efficient way to protect phages from UV radiation, e.g., the usage of natural extracts and astaxanthin provides some protection against UV exposure (1 mW/cm^2^), but only for up to 5 min [183]. Due to the above-mentioned reasons, more research is required to acquire phage stability for storage and preservation without hampering their activity.

This review presents an update on the state of the art of bacteriophage stabilization, including the research published only last year (since May 2021). The review is mostly focused on polymer-based stabilization, encapsulation, lyophilization, and nano-assisted solutions; the problems and future perspectives of such approaches are also highlighted.

## Figures and Tables

**Figure 2 pharmaceutics-14-01936-f002:**
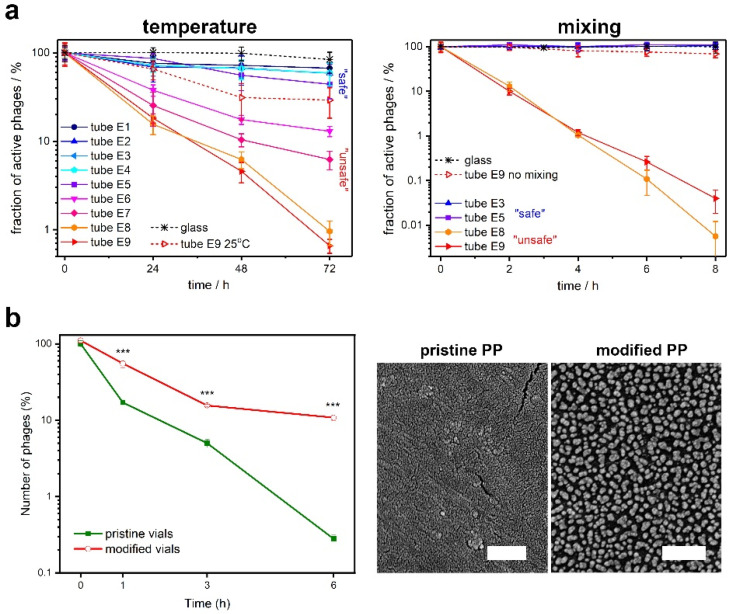
The absorption of bacteriophages on plastic surfaces. (**a**) A comparison of absorption of phages on different types of plastic surfaces enhanced by temperature and mixing [95]. (**b**) The reduction in phage absorption on polypropylene vial surface upon covering with gold-polyoxoborates nanocomposite (*** *p* < 0.001) [97]. Panels were adapted from Richter et al. [95], and Wdowiak et al. [97], based on the CC BY 4.0 License.

**Figure 3 pharmaceutics-14-01936-f003:**
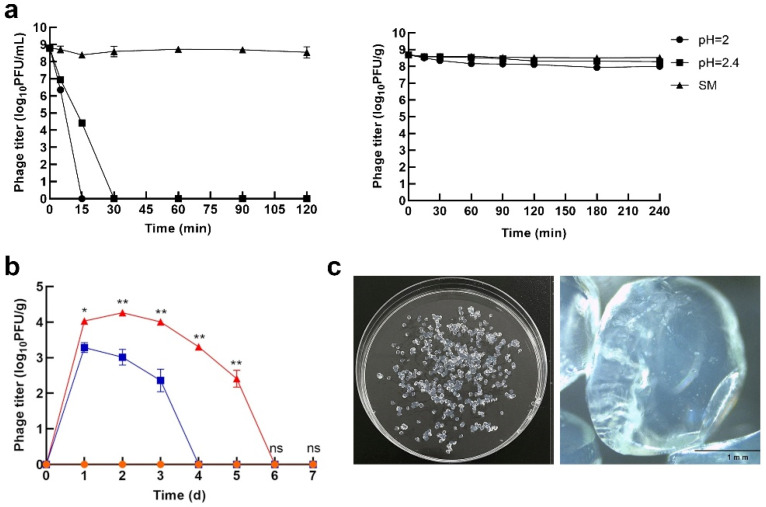
Microencapsulated phages show prolonged stability in gastrointestinal environments and high therapeutic efficiency in treating Escherichia coli O157:H7 infection. (**a**)—the comparison of free phages’ and microencapsulated phages’ stability in simulated gastrointestinal conditions. Encapsulated phages remained active two times longer (30 min) than free phages (15 min). (**b**)—Phage titer in the feces; encapsulated phages remained active for 6 days, while free phages were detected up to 4th day (* *p* < 0.05; ** *p* < 0.01). (**c**)—Images of microcapsules on the plate and imagined using the optical microscope [108]. Panels were adapted from Yin et al. [108], based on the CC BY 4.0 License.

**Figure 4 pharmaceutics-14-01936-f004:**
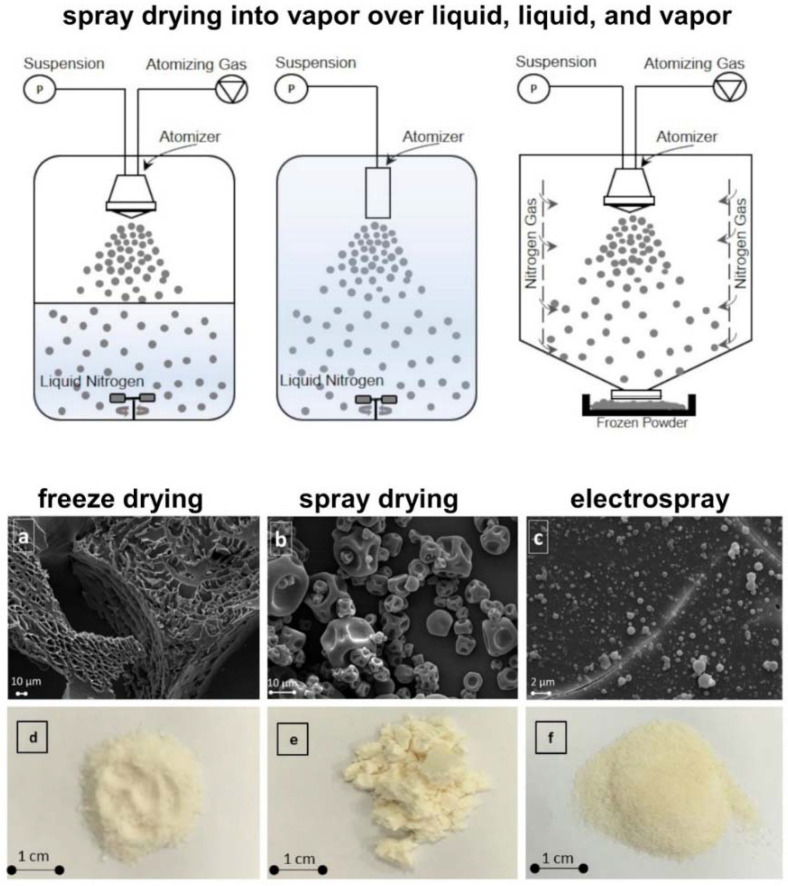
The schematic illustration of spray-drying protocols in vapor over liquid, liquid, and vapor phases(upper, adapted from Adali et al. [114], based on the CC BY 4.0 License), and the comparison of phages lyophilized via freeze-drying (**a**,**d**), spray-drying (**b**,**e**), and electrospray (**c**,**f**). Both SEM images and phage powder form show significant differences in phage formulations (adapted from Ergin et al. [115], based on Elsevier License No. 5375361146341).

**Figure 5 pharmaceutics-14-01936-f005:**
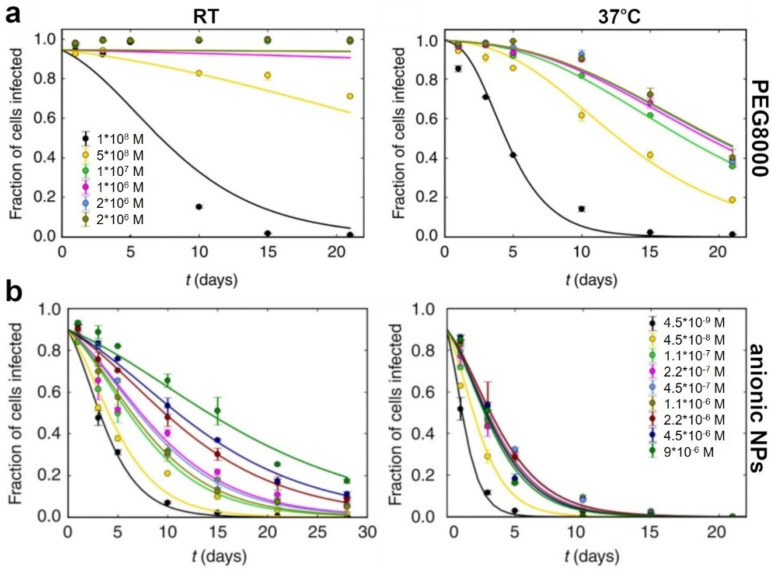
The efficacy of PEG8000 and anionic nanoparticles (NPs) as the thermal-protective additives in adenovirus-based vaccines. (**a**)—Fraction of cells infected versus storage time for Ad5 stored in the presence of different concentrations of PEG at RT and at 37 °C, (**b**)—Fraction of cells infected versus storage time in days for Ad5 stored in presence of different concentrations of anionic MUS:OT NPs at RT and at 37 °C. The panel was adapted from Pelliccia et al. [160], based on the CC BY 4.0 License.

**Table 1 pharmaceutics-14-01936-t001:** Examples of formulations for phage stability.

Technique	Stability Enhancers	Phage	Duration	Reference
Polymer-based	Pullulan and Trehalose	*LISTEX P100*	60 days	[87]
	Polyethylene glycol (PEG)	Felix-O1, A511	24 h within bloodstream	[92]
	Polyoxoborate composite	T4, MS2, M13	6 h of incubation	[97]
	Encapsulation	phiIPLA88, phiIPLA35, phiIPLA-RODI, phiIPLA-C1C	2 months	[99]
	Liposome	KØ1, KØ2, KØ3, KØ4, KØ5	48 h within bloodstream	[102]
Freeze-dried powder	Lactose, trehalose, sucrose	*ISP*	37 months	[121]
	Lecithin	PEV2, PEV20	250 days	[130]
Spray-dried powder	Trehalose, dextran, lactose	*LUZ19, Romulus*	12 min of atomizing	[127]
Freezing inside bacterial cells	Glycerol	*Tailed-phages*	~10 months	[151]
Nano-assisted	Nanolayers of aluminum oxide	λ	~1 month	[159]
	Carbon-based nanoparticles, gold nanoparticles	M13	16 days	[162]

## Data Availability

Not applicable.

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
