# Peer review of "Enhancing the Stability of Bacteriophages Using Physical, Chemical, and Nano-Based Approaches: A Review"

_pharmaceutics, 2022, doi:10.3390/pharmaceutics14091936_

Round 1
Reviewer 1 Report
The present manuscript entitled “Enhancing the stability of bacteriophages by physical, chemical, and nano-based approaches: a review” has described various methods related to the stability and storage of phages. The topic and content of the review are not new in the field of bacteriophage therapy, we can consider this an updated summary. Specific comments are as follows;
1. Although there is less information about bacteriophage stability in the introduction, it is very well-defined and cleverly organized from the phage's initial origin to its classification, host activity, multiplication, and marketing products. Additionally, they can include a few sentences about how the FDA recently approved phage therapy in India during COVID, current updates on the Sciences and regulation of bacteriophage by the US FDA and NIAID, and the present scenario of clinical trials of phage products.
2. No explanations are written to describe the inside of figures 3, (3a, 3b, and 3c) on page no. 7, which is just pasted there for the purpose of keeping the image.
3. Does the statement " Also, nano-based solution for phage stabilization is extremely scarce" in the conclusion (Page No. 11, Line No. 433) mean that nano-solutions or nano-based formulations are not a good choice for phage stability because it would be in conflict with the statement above explained, " Nano-assisted stabilization" (Page No. 9, Line 359-365)? If so, please provide clarification with examples or references.
4. Authors can add a table summarizing the methods of stability of bacteriophages formulations with their merit and demerit as presented in the tables of the following articles-
1. Rosner D, Clark J. Formulations for Bacteriophage Therapy and the Potential Uses of Immobilization. Pharmaceuticals (Basel). 2021 Apr 13;14(4):359. doi: 10.3390/ph14040359. PMID: 33924739; PMCID: PMC8069877.
2. Vandenheuvel, D., Lavigne, R. and Brüssow, H., 2015. Bacteriophage therapy: advances in formulation strategies and human clinical trials. Annual review of virology, 2, pp.599-618.
5. Can the authors explain how this review is novel?
Reviewer 2 Report
The review article by Wdowiak et al. on "Enhancing the stability of bacteriophages by physical, chemical and nano-based approaches' is one of the interesting topics to discuss in phage research. Though the article is completely a review of literature rather than a discussion of new ideas. The article is well written and presented.
Comments:
1. Line no. 42-43: Either omit the classification or report the newest ICTV classification.
2. Line no. 107-113; 123-130; figure 1: Not related to this review.
3. Line no. 169: Give examples for synthetic.
4. Line no. 231: Discuss how encapsulation (encapsulated phages) can be used for transportation.
5. Line no. 341: A table would be better to explain phage storage.
Round 2
Reviewer 1 Report
Responses are satisfactory.
Author Response
Thank You for Your kind suggestions.